EMBO
Molecular Medicine

# Faecal microbiota transplantation protects against radiation-induced toxicity

Ming Cui[1,*,†] iD, Huiwen Xiao[1,†], Yuan Li[1], Lixin Zhou[2], Shuyi Zhao[1], Dan Luo[1], Qisheng Zheng[1], Jiali Dong[1], Yu Zhao[1], Xin Zhang[1], Junling Zhang[1], Lu Lu[1], Haichao Wang[1,3] & Saijun Fan[1,**] iD

## Abstract

Severe radiation exposure may cause acute radiation syndrome, a possibly fatal condition requiring effective therapy. Gut microbiota can be manipulated to fight against many diseases. We explored whether intestinal microbe transplantation could alleviate radiation-induced toxicity. High-throughput sequencing showed that gastrointestinal bacterial community composition differed between male and female mice and was associated with susceptibility to radiation toxicity. Faecal microbiota transplantation (FMT) increased the survival rate of irradiated animals, elevated peripheral white blood cell counts and improved gastrointestinal tract function and intestinal epithelial integrity in irradiated male and female mice. FMT preserved the intestinal bacterial composition and retained mRNA and long non-coding RNA expression profiles of host small intestines in a sex-specific fashion. Despite promoting angiogenesis, sex-matched FMT did not accelerate the proliferation of cancer cells in vivo. FMT might serve as a therapeutic to mitigate radiation-induced toxicity and improve the prognosis of tumour patients after radiotherapy.

**Keywords** faecal microbiota transplantation; gastrointestinal toxicity; gut microbiota; radiation syndrome; radiotherapy

**Subject Categories** Cancer; Digestive System; Microbiology, Virology & Host Pathogen Interaction

## Introduction

Radiation exposure in a mass casualty setting is a serious military and public health concern (Taniguchi *et al*, 2014). Exposure to a high dose of irradiation in a short time is associated with bone marrow toxicity (haematopoietic syndrome) and gastrointestinal (GI) toxicity (GI syndrome), which are collectively known as acute radiation syndrome (ARS; Kirsch *et al*, 2010; Leibowitz *et al*, 2014). ARS may facilitate an intractable pathologic process and even cause eventual death (Lee *et al*, 2014). In addition, mounting clinical evidence has shown the bone marrow and small intestine epithelium to be the major sites of injury during radiation therapy, owing to their higher sensitivity to ionizing radiation (Ciorba *et al*, 2012). Radiation-mediated toxicity, especially radiation-induced gastrointestinal injury, is a medical problem that urgently needs effective therapy.

The GI tract has the unique property of harbouring numerous microbes within the lumen, most of which are bacteria that have co-evolved with the host in a mutualistic relationship (Kamada *et al*, 2013). Recently, investigations focusing on gut microbiota have experienced a renaissance, and growing evidence supports a pivotal role of intestinal microbes as key regulatory elements in their host's physiologic and pathologic status (Ostaff *et al*, 2013). For example, intestinal microbes govern metabolic function and energy balance, and flora disequilibrium is deemed to contribute to the development of numerous metabolic diseases (Amar *et al*, 2011; Nieuwdorp *et al*, 2014). Intestinal microbes are continuously shaping the development of their host's immune system, directly modulating the innate and adaptive immune responses (Sommer & Backhed, 2013; Barroso-Batista *et al*, 2015). Epidemiological studies reveal that the intestinal microbiome engages in the pathogenesis of inflammatory bowel disease (IBD) with characteristic shifts in the composition of the intestinal microbiota, reinforcing the view that IBD results from altered interactions between intestinal microbes and the mucosal immune system (Kostic *et al*, 2014). Germ-free mice and faecal transplant research demonstrate that changes in the microbiota are necessary and sufficient for both low-grade inflammation and metabolic syndrome (Chassaing *et al*, 2015). Moreover, marked shifts in bacterial communities in the gut are inextricably intertwined with the development of diet-associated cancer (Schulz *et al*, 2014; Feng *et al*, 2015; Gorjifard & Goldszmid, 2015), suggesting that enteral bacteria might be used as diagnostic biomarkers for many cancers. However, the relationship between

1 Tianjin Key Laboratory of Radiation Medicine and Molecular Nuclear Medicine, Institute of Radiation Medicine, Chinese Academy of Medical Sciences and Peking Union Medical College, Tianjin, China
2 Key Laboratory of Carcinogenesis and Translational Research (Ministry of Education/Beijing), Department of Pathology, Peking University Cancer Hospital & Institute, Beijing, China
3 Department of Emergency Medicine, North Shore University Hospital, Laboratory of Emergency Medicine, the Feinstein Institute for Medical Research, Manhasset, NY, USA
*Corresponding author. Tel: +86 022 85685301; E-mail: cuiming0403@bjmu.edu.cn
**Corresponding author. Tel: +86 022 85685301; E-mail: fansaijun@irm-cams.ac.cn
†These authors contributed equally to this work

enteric bacteria and the radiosensitivity of hosts is unknown; moreover, whether intestinal microbes can be used to mitigate radiation-induced injury remains undocumented.

In the present study, we evaluated the therapeutic potential and molecular mechanisms of faecal microbiota transplantation (FMT) in a rodent model of lethal irradiation and demonstrated that transplantation of enteric microbes from healthy mice remarkably mitigated radiation-induced GI syndrome in a sex-dependent fashion. More importantly, FMT retained the mRNA and long non-coding RNA (lncRNA) expression profiles of the small intestine. Thus, our findings provide new insights into the function and underlying protective mechanism of intestinal microbiota transplant in the

context of radiation-induced toxicity in a preclinical experimental setting.

# Results

## Intestinal bacterial communities are associated with radiosensitivity in a mouse model

To determine the relationship between the intestinal bacterial pattern and radiosusceptibility, we used 16S rRNA sequencing to analyse the enteric bacterial profile in the mice faeces after

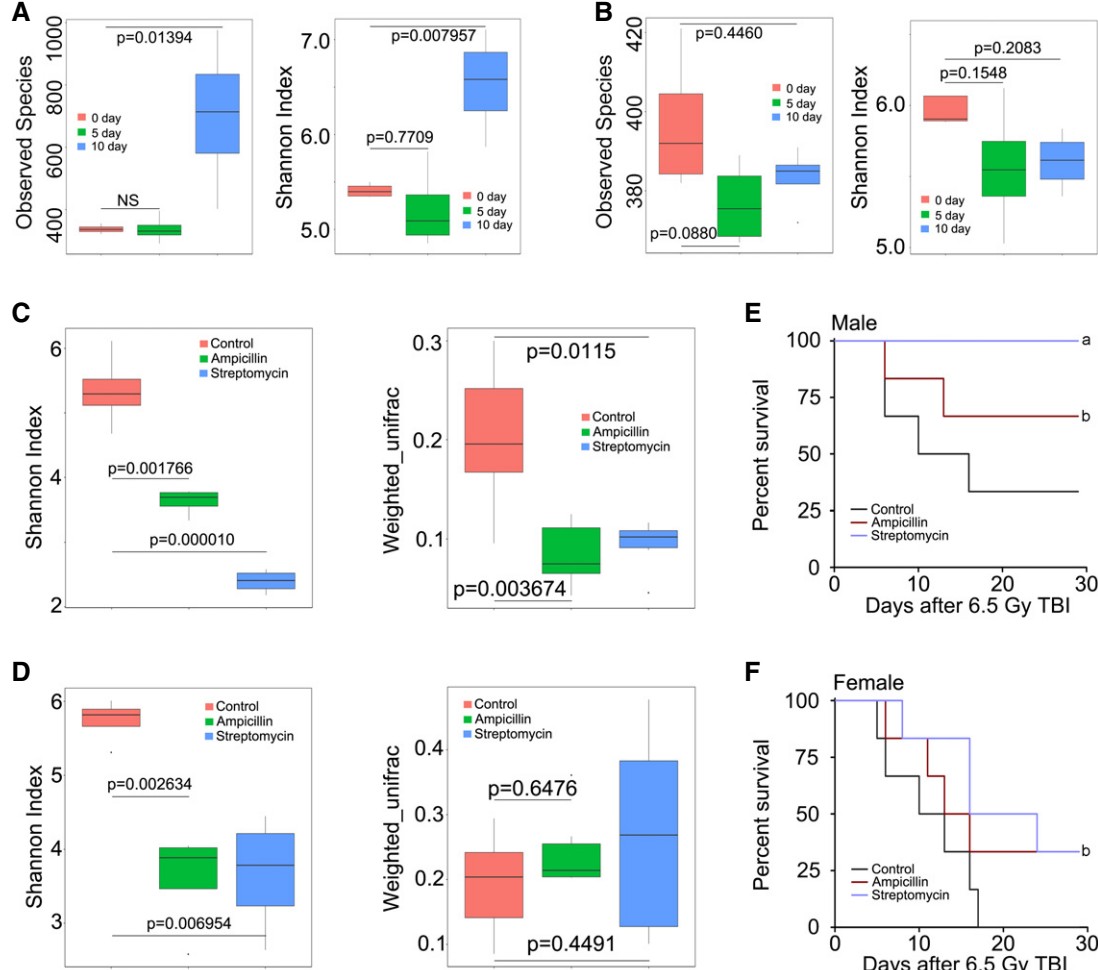

**Figure 1. The intestinal bacterial flora profile is associated with radiosensitivity in a mouse model.**

A, B   The observed species number and Shannon diversity index of intestinal bacteria in male (A) and female (B) mice were examined by 16S high-throughput sequencing after irradiation at days 5 and 10, $n = 4$ per group. Statistically significant differences are indicated: Wilcoxon rank sum test. The top and bottom boundaries of each box indicate the 75th and 25th quartile values, respectively, and lines within each box represent the 50th quartile (median) values. Ends of whiskers mark the lowest and highest diversity values in each instance.

C, D   The Shannon diversity index and β-diversity of intestinal bacteria in male (C) and female (D) mice were examined by 16S high-throughput sequencing after 6 weeks of antibiotics treatment, $n = 4$ per group. Statistically significant differences are indicated: Wilcoxon rank sum test. The top and bottom boundaries of each box indicate the 75th and 25th quartile values, respectively, and lines within each box represent the 50th quartile (median) values. Ends of whiskers mark the lowest and highest diversity values in each instance.

E, F   Kaplan–Meier survival analysis of saline-treated, ampicillin-treated, and streptomycin-treated male (E) and female (F) mice after 6.5 Gy TBI were performed ([a]$P < 0.05$ by log-rank test between antibiotic-treated and saline-treated groups; [b]$P < 0.005$ by log-rank test between antibiotic-treated and saline-treated groups, $n = 6$ per group).

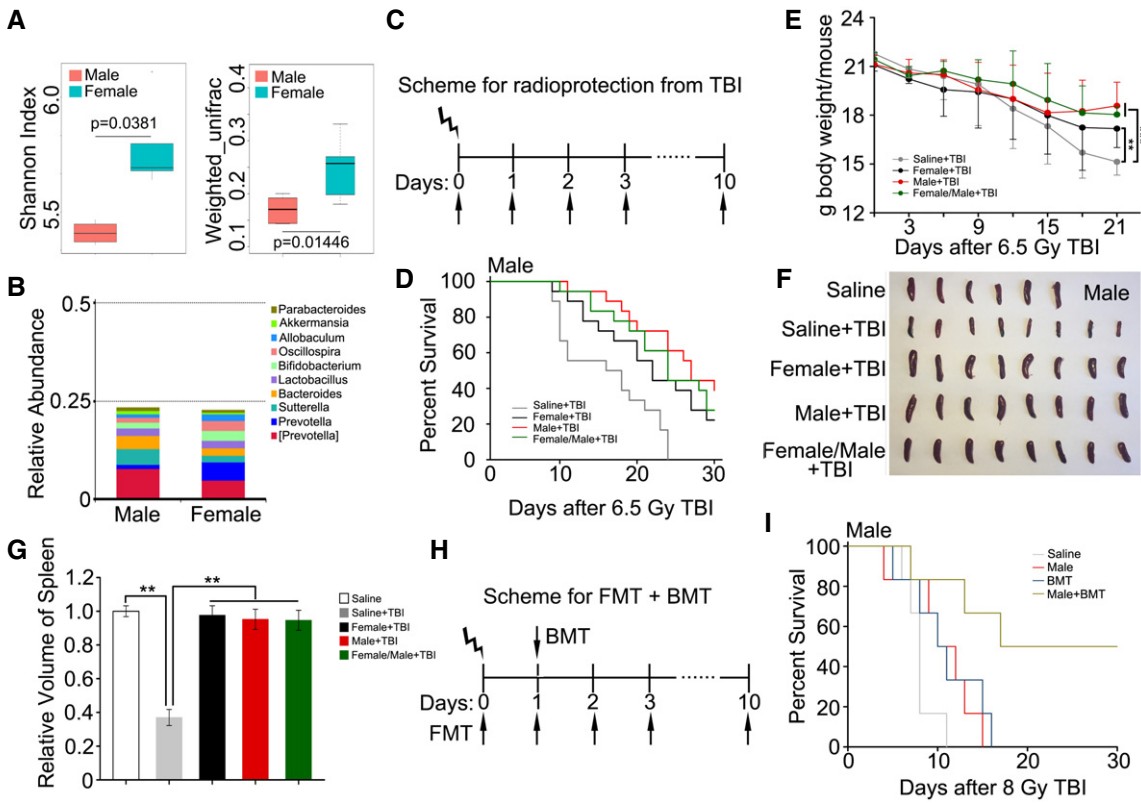

**Figure 2. Gavage of faecal microbiota protects against radiation-induced death and haematopoietic toxicity.**

A    The Shannon diversity index and β-diversity of intestinal bacteria between male and female mice without irradiation were assessed by 16S high-throughput sequencing, *n* = 4. Statistically significant differences are indicated: Student's *t*-test. The top and bottom boundaries of each box indicate the 75th and 25th quartile values, respectively, and lines within each box represent the 50th quartile (median) values. Ends of whiskers mark the lowest and highest diversity values in each instance.

B    The relative abundances of the top 10 bacteria at the genus level in male and female mice were assessed using 16S high-throughput sequencing, *n* = 4.

C    Scheme for radioprotection after TBI.

D    Male mice received 10 days of oral gavage with saline, male gut microbes, female gut microbes or a male/female gut microbe mixture after 6.5 Gy TBI, Kaplan–Meier survival analysis of the mice was performed. *P* < 0.005 by log-rank test between the saline + TBI group and the male + TBI group, *P* < 0.05 by the log-rank test between the saline + TBI group and the female + TBI group (or female/male + TBI group), *n* = 18 per group.

E    Male mice received 10 days of oral gavage with saline, male gut microbes, female gut microbes or a male/female gut microbe mixture after 6.5 Gy TBI, and the body weight was measured. Mean ± SD. Significant differences are shown relative to Saline + TBI group: **P* < 0.01, ***P* < 0.001; Student's *t*-test, *n* = 18 per group.

F, G    Photographs (F) and volume (G) of dissected spleens from radiation-exposed male mice, the spleens were obtained at day 15 after 6.5 Gy TBI. Mean ± SD. Significant differences are shown relative to Saline + TBI group: **P* < 0.01; Student's *t*-test, *n* = 6 in Saline group, *n* = 8 in experimental groups.

H    Scheme for faecal microbiota transplantation combined with bone marrow transplant after TBI.

I    Kaplan–Meier survival analysis of the irradiated mice treated with saline, FMT, BMT or FMT combined with BMT was performed. *P* < 0.005 by log-rank test between FMT combined with BMT and other three groups, *n* = 12 per group.

irradiation exposure on days 5 and 10. As shown in Fig 1A and B, 6.5 Gy gamma ray altered the diversity and composition of enteric bacteria in male but not in female C57BL/6 mice. Antibiotic treatment has been reported to be capable of shifting the intestinal bacterial communities in mice (Theriot *et al*, 2014; Brown *et al*, 2016). Accordingly, mice were administered drinking water containing ampicillin or streptomycin for 6 weeks, and then, we performed 16S rRNA sequencing analysis and obtained that enteric bacterial composition of antibiotic-fed mice (Abt mice) was quite different from that of controls (drinking water without antibiotics; Fig 1C and D); however, the gut microbiota of control mice did not change overtly during the 6 weeks (Appendix Fig S1A and B). After 6.5 Gy total body irradiation (TBI), our observations revealed that the survival rate of Abt mice was significantly higher than that of controls, indicating that changes in the intestinal bacterial

communities are able to influence the radiosensitivity of both male and female mice (Fig 1E and F). A decrease in peripheral white blood cell (WBC) counts has been observed in irradiated animals (Deng *et al*, 2015). Although we observed that ampicillin and streptomycin conferred protection against sub-lethal irradiation, these antibiotics did not heighten peripheral WBC counts (Appendix Fig S1C and D), indicating that antibiotics could not ameliorate radiation-induced haematopoietic syndrome.

## Gavage of faecal microbiota protects against radiation-induced death

Divergent factors, such as genetic predisposition, diet and inflammation states, can differently affect enteric bacterial flora (Smith *et al*, 2015; Schaubeck *et al*, 2016). Therefore, we compared the diversity

of gut bacterial composition between male and female C57BL/6 mice using 16S rRNA sequencing before irradiation. Interestingly, we found that the bacteria taxonomic proportions in stool of male and female mice were different (Fig 2A). Compared with female mice, male animals harboured a higher frequency of *Bacteroides* but a lower frequency of *Prevotella* at the genus level (Fig 2B and Appendix Fig S1E). Accordingly, we performed sex-matched or mismatched FMT according to the following scheme: female faecal microbiota transplants to male or female mice (female + TBI); male faecal microbiota transplants to male or female mice (male + TBI); and a mixture of faecal microbiota from male and female transplants to male or female mice (female/male + TBI, the weight of male and female stool was 1:1); saline was used as a control (saline + TBI). The C56BL/6 mice were separated into their respective cohorts and gavage with faecal microbiota or saline after TBI, as depicted in Fig 2C. Notably, the faecal microbiota treatments overtly increased the survival rate and body weight of male and female mice after 6.5 Gy TBI (Figs 2D and E, and EV1A and B). In particular, the optimal results were obtained when the gender of donors matched with that of recipients, indicating that FMT is an effective therapy against radiation-induced death in a mouse model and its efficiency is determined by the gender matched between the donor and recipient. We also repeated the experiments using older donors, but the three kinds of FMT all failed to increase the survival rate of irradiated male and female mice (data not shown).

To better understand the effect of the gut flora on the haematopoietic system, we checked the size of the spleens from each group. The size of the spleens was reduced after irradiation, but restored in the female + TBI, male + TBI and female/male + TBI treatment groups (Figs 2F and G, and EV1C and D). Moreover, these treatment regimens also elevated WBC counts slightly in the peripheral blood (Fig EV1E and G) without affecting blood haemoglobin levels (Fig EV1F and H). Because TBI ablates all marrow and FMT only mitigates TBI-induced haematopoietic toxicity slightly, we further performed FMT combined with bone marrow transplant (BMT; Fig 2H). After 8 Gy irradiation exposure, the survival rate of mice treated with FMT combined with BMT was much higher than that of mice treated FMT or BMT alone (Figs 2I and EV1I), suggesting that FMT combined with BMT might significantly mitigate irradiation-induced toxicity.

## FMT restores GI tract function and epithelial integrity after irradiation

On the basis of the aforementioned observations, we further untangled the impact of FMT on GI function and integrity in animals receiving faecal microbiota transferred from the same gender after irradiation exposure. Day 21 after 6.5 Gy TBI, haematoxylin and eosin (H&E) staining revealed a dramatic decrease in the number of intact intestinal villi, which was rescued by gut microbiota treatment (Figs 3A and EV2A). The Alcian Blue periodic acid-Schiff (AB-PAS) and periodic acid-Schiff (PAS) staining further showed that FMT thickened the mucus layer and increased the number of goblet cells obviously which injured by irradiation exposure (Figs 3A and EV2A) in male and female mice. In addition, the total amount of formed stool gathered from the cage of the saline-treated group was much lower than that of the FMT group (Figs 3B and C, and EV2B and C). Moreover, FMT decreased the radiation-heightened FITC–

dextran level in peripheral blood (Figs 3D and EV2D), suggesting that FMT improves GI tract function and epithelial integrity in irradiated animals. Using quantitative polymerase chain reaction (qPCR), we further validated the finding that the expression of *Muc2*, *Glut1* (*Slc2a1*), *Pgk1*, intestinal trefoil factor (*TFF3/ITF1*) and multidrug resistance protein 1 (*MDR1*) (Figs 3E–I and EV2E–I), which all participate in epithelial integrity maintaining after toxic stimuli, reached about threefold higher levels in the small intestine tissues of both irradiated male and female mice after FMT. Together, our observations demonstrate that gavage of gut microbiota improves GI tract function and epithelial integrity after irradiation.

## FMT alters gut bacterial composition structure after irradiation

To elucidate the mechanism of FMT mitigating radiation-induced toxicity, we performed 16S rRNA sequencing to analyse the bacterial taxonomic composition in faecal microbiota-treated and saline-treated animals. Irradiation caused persistent dysbiosis in the gut bacterial ecosystem at days 5 and 10, and FMT effectively erased these changes; in male mice, for example, irradiation continually decreased the frequency of *Bacteroides* (or *Lactobacillus*) at the genus level (Fig 4A), but male + TBI treatment increased (or stabilized) the frequency of *Bacteroides* (or *Lactobacillus*) at day 5 (Fig 4B). In contrast, female + TBI treatment led to a reduction in *Bacteroides* and a bloom in *Lactobacillus* at day 5, whereas female/male + TBI treatment led to opposite alterations in the flora (Fig EV3A and B). At day 10, the frequency of *Bacteroides* and *Lactobacillus* dropped in all gut microbe-treated male mice (Figs 4A and B, and EV3A and B). Moreover, in the saline and female treatment groups, the frequency of *Prevotella* at the genus level decreased progressively (Figs 4C and EV3C), whereas the frequency of *Prevotella* was increased in male and female/male treatment cohorts (Figs 4D and EV3D). In female mice, the frequency of *Bacteroides* was elevated at day 5 and then declined to a lower level at day 10 in almost all animals (Figs 4E and F, and EV3E and F). Notably, female + TBI treatment recovered the frequency of *Bacteroides* to the level of controls at day 10 (Fig 4F). In addition, at day 10, the saline, male and female/male treatment groups showed a reduction in the *Prevotella* frequency at the genus level (Figs 4G and EV3G and H), but the female treatment group exhibited the reverse outcome (Fig 4H). Thus, our observation indicates that FMT has a profound influence on the radiation-altered gut bacterial composition.

Next, principal component analysis (PCA) was used to further determine the role of FMT in changing the intestinal bacterial flora profile. As shown in Fig 5A, the gut bacterial composition profile substantially changed after radiation exposure at day 10 in male mice and was inhibited by gut microbial therapy (Figs 5B and EV4A and B). As expected, irradiation exposure shaped the intestinal bacterial composition of female mice, and microbial therapy altered the composition of faecal microbial flora in these animals (Figs 5C and D, and EV4C and D). Specifically, irradiation exposure caused a down-regulation of the relative abundance of Bacteroidetes (or Firmicutes) at the phylum level in male mice (or in female mice) at day 10, but FMT reversed this down-regulation (Figs 5E–H and EV4E–H). Together, our observation demonstrates that gut microbiota treatment preserves the gut bacterial composition in both male and female mice after exposure to radiation.

                                          

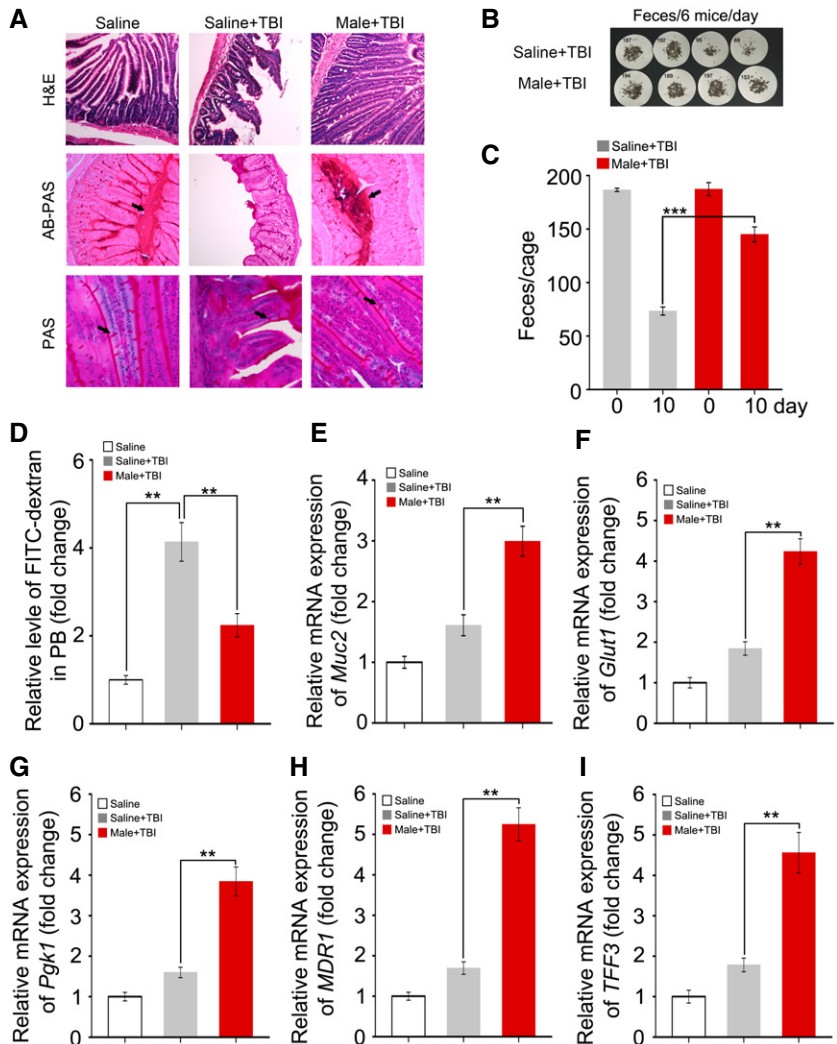

**Figure 3.  FMT ameliorates GI tract function and epithelial integrity after irradiation.**

Male mice were separated into two groups after 6.5 Gy gamma ray exposure, where one cohort was treated with saline as a control and the other was treated with sex-matched FMT.

A   The morphology of the small intestine in the radiation-induced mice treated with saline or sex-matched FMT was shown by H&E, AB-PAS and PAS staining; the small intestine tissues were obtained at day 21 after TBI. The arrows point to the mucus layer or goblet cells.

B, C   Faecal pellet counts removed from cage bedding on 3 day from representative cages is shown. Mean ± SD, *n* = 6 mice per treatment, ***P* < 0.001 by Student's *t*-test between Saline + TBI and Male + TBI group.

D   The FITC–dextran in peripheral blood from saline-treated and sex-matched FMT mice was assessed at day 21 after irradiation exposure. Mean ± SD. Significant differences are indicated: ***P* < 0.01; Student's *t*-test, *n* = 6 per group.

E–I   The expression levels of *Muc2*, *Glut1*, *Pgk1*, *MDR1* and *TFF3* were examined in small intestine tissues from saline-treated and sex-matched FMT mice by quantitative PCR; the small intestine tissues were obtained at day 21 after TBI. Mean ± SD. Significant differences are indicated: ***P* < 0.01; Student's *t*-test, *n* = 12 per group.

## FMT retains the gene expression profile of the small intestine after irradiation

To further decipher the protective mechanism of FMT, we performed gene ontology (GO) analysis using small intestine tissues from saline and sex-matched FMT treatment groups. Interestingly, compared with the saline treatment group, the top 30 terms exhibiting significant expression alterations were mostly involved in innate and adaptive immunity in male mice treated with sex-matched FMT (Fig 6A). However, sex-matched FMT altered the expression of genes involved in metabolism in irradiated female mice, including genes required for cellular lipid catabolic processes and fatty acid beta-oxidation (Fig 6B), suggesting that different sex-specific intestinal microbiota dictates their divergent physiologic status. In addition, we assessed the mRNA expression profile of genes involved in immunity and metabolism by cluster analysis and found remarkable changes in the immune and metabolic gene expression profiles between the gut microbe-treated experimental group and the saline-treated group (Fig 6C–F), indicating that the hosts are responsive to intestinal microbiota treatment. We also compared the gene expression profile of small intestine tissues from irradiated mice

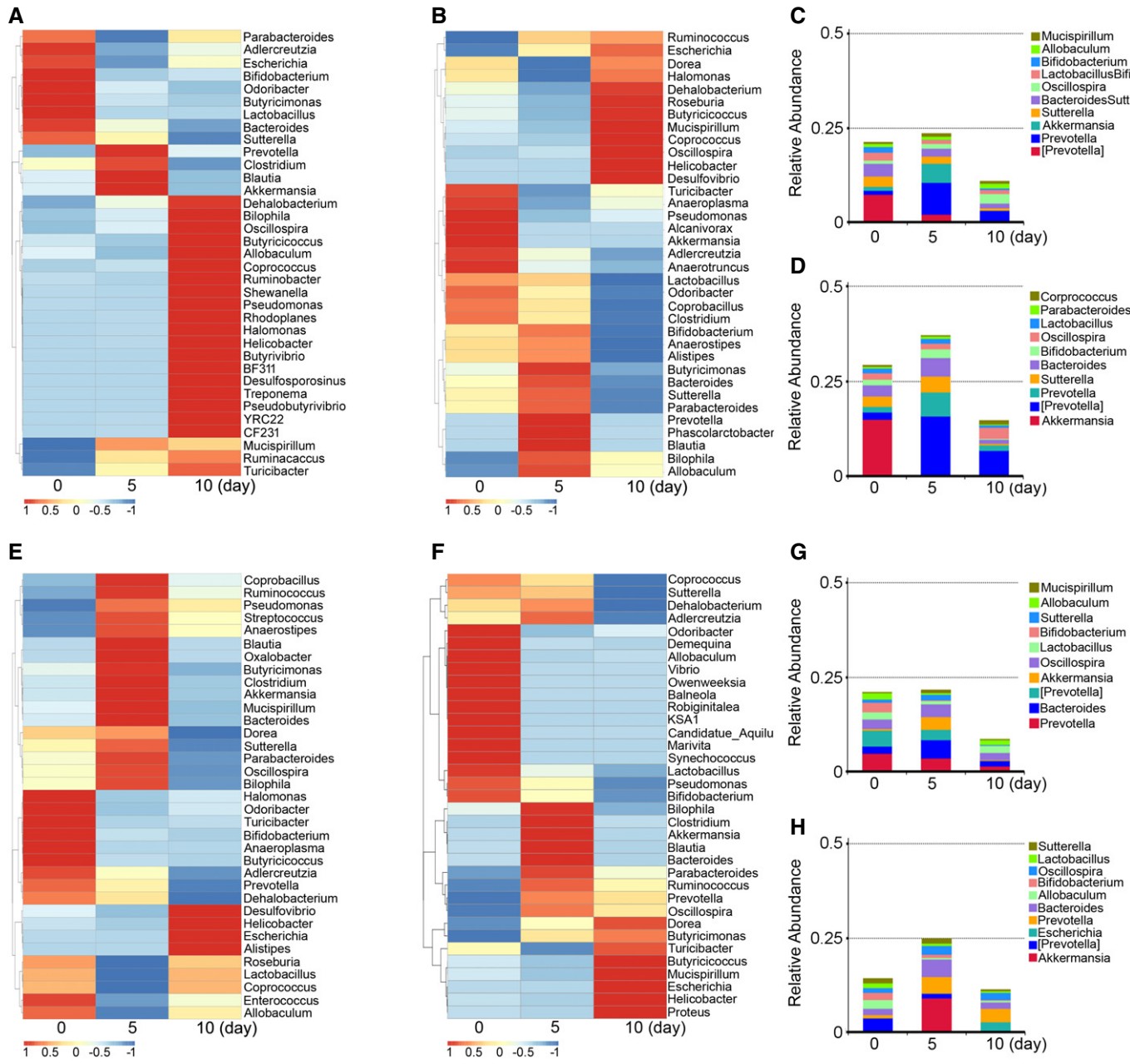

**Figure 4. FMT alters the gut bacterial composition profile after irradiation.**

A, B   The alteration of intestinal bacterial patterns at the genus level in saline-treated (A) and sex-matched FMT (B) male mice was assessed using 16S high-throughput sequencing after irradiation at days 5 and 10, *n* = 4. The heat map is colour-based on row *Z*-scores. The mice with the highest and lowest bacterial level are in red and blue, respectively.

C, D   The relative abundances of the top 10 bacteria at the genus level in saline-treated (C) and sex-matched FMT (D) male mice were assessed using 16S high-throughput sequencing after irradiation at days 5 and 10, *n* = 4.

E, F   The alterations of intestinal bacterial patterns at the genus level in saline-treated (E) and sex-matched FMT (F) female mice were assessed using 16S high-throughput sequencing after irradiation at days 5 and 10, *n* = 4. The heat map is colour-based on row *Z*-scores. The mice with the highest and lowest bacterial level are in red and blue, respectively.

G, H   The relative abundances of the top 10 bacteria at the genus level in saline-treated (G) and sex-matched FMT (H) female mice were assessed using 16S high-throughput sequencing after irradiation at days 5 and 10, *n* = 4.

between saline treatment and sex-mismatched (or sex-mixed) groups. GO analysis revealed that sex-mismatched (or sex-mixed) FMT-enriched top 30 terms were quite different from those

altered by sex-matched FMT in male (Appendix Fig S2A–D) and female mice (Appendix Fig S3A–D), which further validated that the effect of sex-specific intestinal microbiota on hosts was

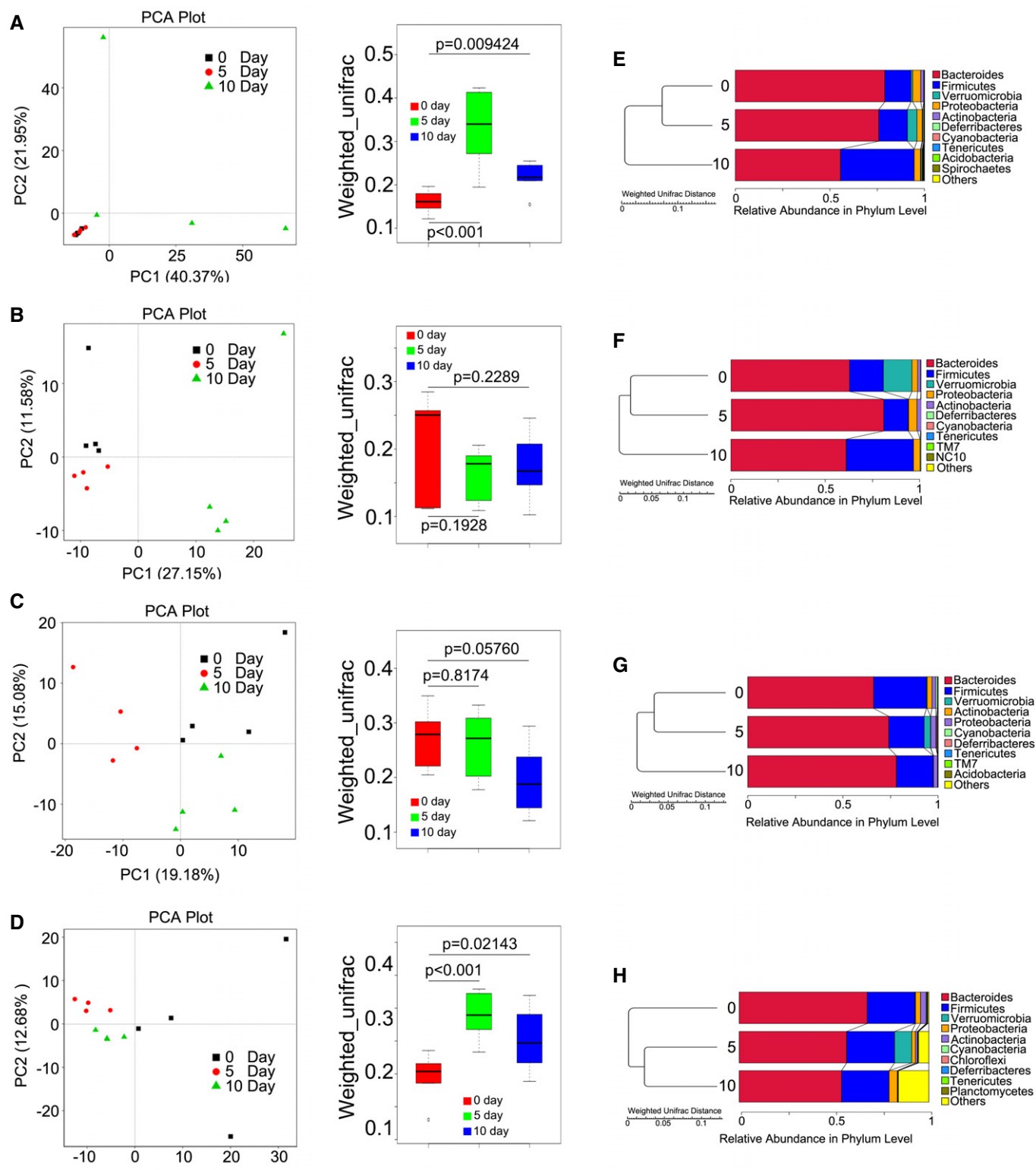

**Figure 5.**

different. Given that lncRNAs have been reported to have an important role in immunity and metabolism modulation, we examined the lncRNA expression patterns related to the mRNAs

in our study and observed that lncRNA expression was retained in sex-matched FMT-treated mice (Appendix Figs S4–S7), indicating that the intestinal microbiota transplant affects the host's

**Figure 5.  FMT changes gut bacterial community structure after irradiation.**

Male and female mice were separated into two groups after 6.5 Gy gamma ray exposure, where one cohort was treated with saline as control and the other was treated with sex-matched FMT.

A, B   Principal component and β-diversity analysis were used to measure the shift of the intestinal bacterial composition profile in saline-treated (A) and sex-matched FMT (B) male mice after irradiation at days 5 and 10, *n* = 4. Statistically significant differences are indicated: Wilcoxon rank sum test. The top and bottom boundaries of each box indicate the 75th and 25th quartile values, respectively, and lines within each box represent the 50th quartile (median) values. Ends of whiskers mark the lowest and highest diversity values in each instance.

C, D   Principal component and β-diversity analysis were used to measure the shift of the intestinal bacterial composition structure in saline-treated (C) and sex-matched FMT (D) female mice after irradiation at days 5 and 10, *n* = 4. Statistically significant differences are indicated: Wilcoxon rank sum test. The top and bottom boundaries of each box indicate the 75th and 25th quartile values, respectively, and lines within each box represent the 50th quartile (median) values. Ends of whiskers mark the lowest and highest diversity values in each instance.

E, F   The relative abundances of enteric bacteria at the phylum level in saline-treated (E) and sex-matched FMT (F) from male mice were assessed using 16S high-throughput sequencing after irradiation at days 5 and 10.

G, H   The relative abundances of enteric bacteria at the phylum level in saline-treated (G) and sex-matched FMT (H) from female mice were assessed using 16S high-throughput sequencing after irradiation at days 5 and 10.

mRNA and lncRNA expression profiles to mitigate radiation-induced toxicity.

## FMT enhances angiogenesis but does not accelerate tumour growth

Angiogenesis has been regarded as a protective effector against radiation-induced toxicity. Thus, we further investigated whether intestinal microbiota transplantation improved angiogenesis in the system. Intriguingly, immunohistochemical (IHC) staining revealed that the expression of F8, a marker for angiogenesis, was elevated in gut microbe-treated mice compared with that of controls (Fig 7A). qPCR assays further validated that gut microbe transplant up-regulated the expression level of *Vegf* mRNA in small intestine (Fig 7B), suggesting that FMT accelerates angiogenesis in the small intestine of irradiated male and female mice.

Angiogenesis also shows a positive relationship with tumour development. Radiotherapy is one of the most successful and widely used cancer therapies (Svensson *et al*, 2006). To investigate whether FMT can be used to improve prognosis in tumour radiotherapy, we further examined whether FMT accelerated tumour cell proliferation *in vivo*. Accordingly, C57BL/6 mice were injected with HT29 (or A549) tumour cells subcutaneously (Fig EV5A and B) and treated with faecal microorganisms for 10 days. We found that gut microbe treatment did not significantly change the animals' body weight (Fig EV5C–F) and the volume and weight of the tumours in animals receiving exogenous cancer cells (Fig 7C–F), suggesting that intestinal microbiota transplant might be employed as a radioprotector to improve prognosis in cancer radiotherapy. Together, our data demonstrate that FMT improves angiogenesis without facilitating tumour growth.

## Discussion

The gastrointestinal tract is inhabited by a dense population of organized and highly specialized microbial flora that collectively modulates host immunity and metabolism. Compositional and functional changes in commensal microbiota are thought to be involved in the pathogenesis of many diseases (Petrof & Khoruts, 2014; Vanhoecke *et al*, 2015). Recently, epidemiological and clinical studies on symbiotic microbiota have experienced a renaissance

(Vasconcelos *et al*, 2016). Understanding how the enteric microbiota affects health and disease requires a paradigm shift from focusing on individual pathogens to an ecological approach that considers the community as a whole (Lozupone *et al*, 2012). These intestinal microbes boost biofilm formation by facilitating microbial co-aggregation and the production of biosurfactants and bacteriocins, which selectively kill other microorganisms to maintain microbiota stability, enhance gut barrier function through interacting with epithelia and modulate the host immune function (Borody & Khoruts, 2012). It is thus possible that pharmacological modification of the intestinal microbiome can be a therapeutic strategy for the treatment of many human diseases (Owyang & Wu, 2014). For example, targeted restoration of the intestinal bacteria is beneficial in treating recalcitrant or recurring *C. difficile* infection (Lawley *et al*, 2012). Strategically, FMT is the most direct and radical way to alter the composition of a human's enteric microbiota to improve the quality of life in patients with inflammatory bowel disease (Moayyedi *et al*, 2015; Wei *et al*, 2015), non-alcoholic fatty liver disease (Le Roy *et al*, 2013), metabolic syndrome (Vrieze *et al*, 2012) and neuropsychiatric disorders (Xu *et al*, 2015). Thus far however, whether FMT could be used as a therapeutic method to ameliorate radiation-induced toxicity remains unknown. Given the changes in the microbiota vary with time, we performed FMT to treat irradiated mice for 10 days and observed that FMT preserved the radiation-impaired enteric bacterial composition during these 10 days. We observed that multiple FMT doses increased the survival rate of irradiated mice suggesting that FMT administration may provide protection against radiation-induced toxicity. Therefore, our findings support the hypothesis that FMT might emerge as a potential therapeutic option for radiation-induced death.

Given that multiple factors, such as age and genetics (Lozupone *et al*, 2012), can drive gut microbiota alteration, the use of FMT should follow guidelines. Clinical studies noted that donors with any gastrointestinal complaints, metabolic syndrome, autoimmune diseases or allergic diseases should be excluded (Borody & Khoruts, 2012). Here, our high-throughput sequencing showed a difference in the intestinal bacterial composition between male and female C57BL/6 mice. Thus, we achieved sex-matched and mismatched FMT and discovered that the optimal therapeutic efficacy of FMT was obtained when the sex of donors matched recipients. Although sex-matched, sex-mismatched and sex-mixed FMT shaped the gut bacterial composition of irradiated mice efficiently, the frequency of

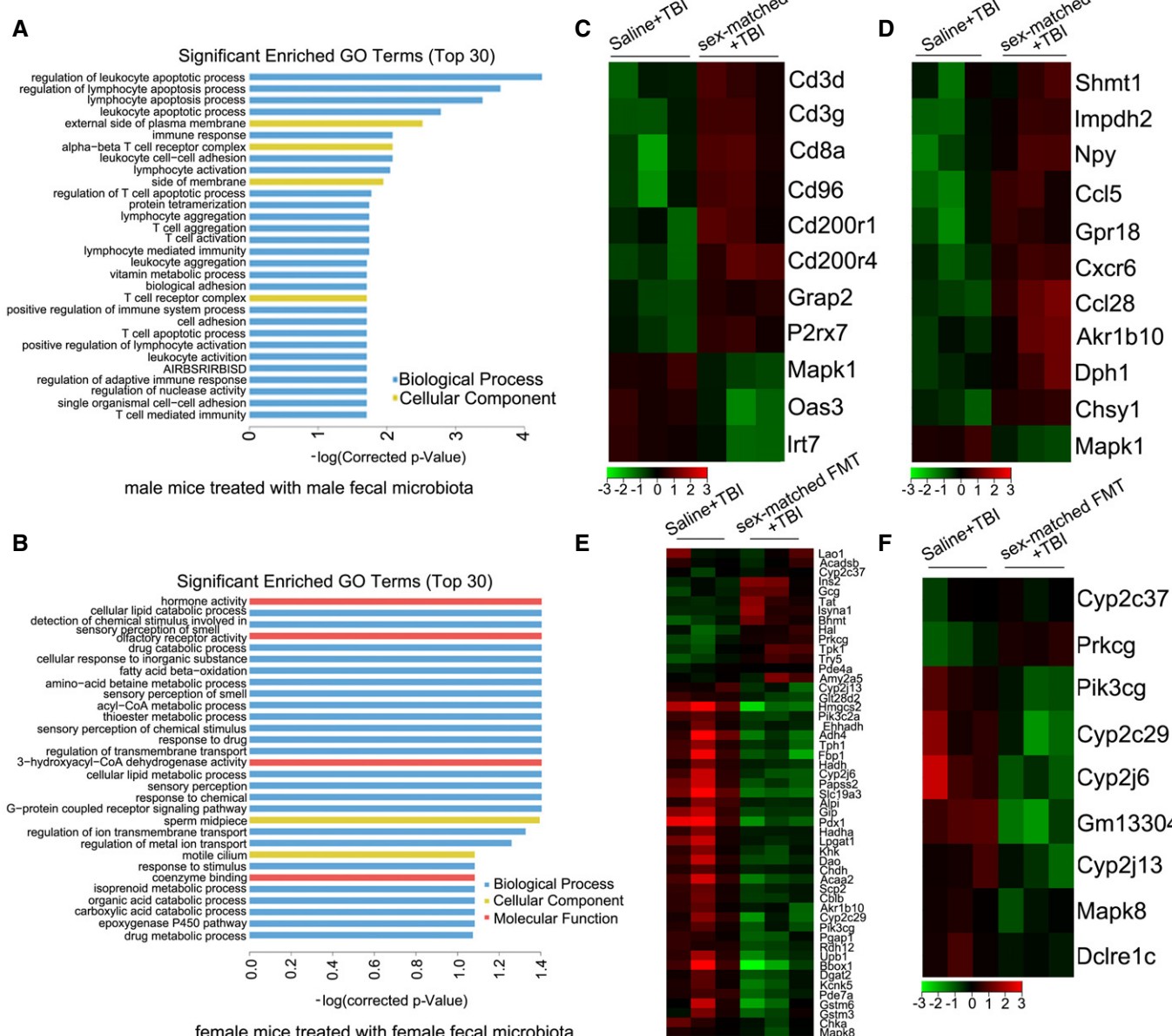

**Figure 6. FMT retains the gene expression profile of the small intestine after irradiation.**

Male and female mice were separated into two groups after 6.5 Gy gamma ray exposure, where one cohort was treated with saline as control and the other was treated with sex-matched FMT. Twenty-one days after irradiation, the mice were euthanized, and the small intestine tissues were excised and microarray analysis was performed.

A    Significantly enriched GO terms (top 30) induced by sex-matched FMT in male mice. AIRBSRIRBISD, adaptive immune response based on somatic recombination of immune receptors built from immunoglobulin superfamily domains.

B    Significantly enriched GO terms (top 30) induced by sex-matched FMT in female mice.

C, D    Heat map of genes significantly up- and down-regulated (red and green, respectively) in small intestine tissue from sex-matched FMT male mice compared with those of controls. The genes were associated with immunity (C) and metabolism (D), and each row represents a single gene.

E, F    Heat map of genes significantly up- and down-regulated (red and green, respectively) in small intestine tissue from sex-matched FMT female mice compared with those of controls. The genes were associated with metabolism (E) and immunity (F), and each row represents a single gene.

specific bacteria changed by the three kinds of FMT was different, suggesting that the colonization of key bacterial strains might contribute to the efficient of FMT-mitigated radiation-induced toxicity. Together, our observations bolster the position that establishing

formal guidelines for FMT is imperative to developing therapies or clinical trials.

Given that the clinical components of acute radiation sickness include intricate haematopoietic syndrome and intractable GI

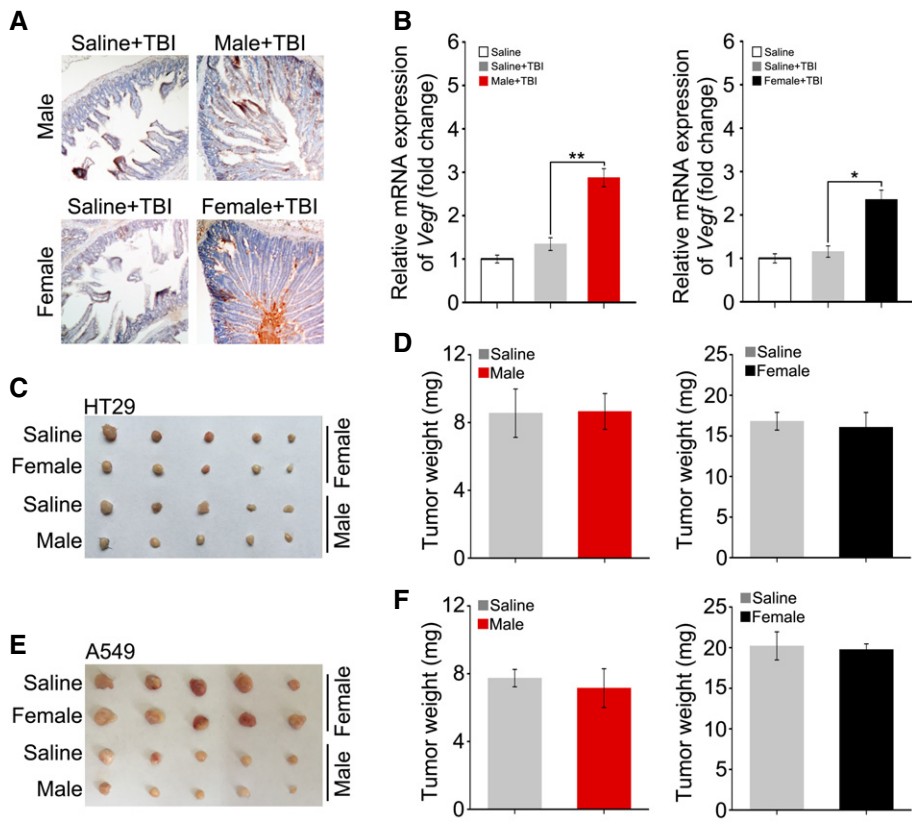

**Figure 7.  FMT enhances angiogenesis without accelerating tumour growth.**

A   The level of F8 was assessed in small intestine tissue by immunohistochemistry in saline-treated and sex-matched FMT male mice (or sex-matched FMT female mice).

B   The expression level of *Vegf* in saline-treated and sex-matched FMT male mice (or sex-matched FMT female mice) was examined by quantitative PCR. Mean ± SD. Significant differences are indicated: *$P < 0.05$, **$P < 0.01$; Student's *t*-test, $n = 12$ per group.

C   Photographs of dissected tumours from saline-treated or sex-matched FMT C57BL/6 mice transplanted with HT29.

D   The weight of tumours from experimental groups of C57BL/6 mice. Mean ± SD. Significant differences are indicated: none significant; Student's *t*-test, $n = 5$ per group.

E   Photographs of dissected tumours from saline-treated or sex-matched FMT C57BL/6 mice transplanted with A549.

F   The weight of tumours from experimental groups of C57BL/6 mice. Mean ± SD. Significant differences are indicated: none significant; Student's *t*-test, $n = 5$ per group.

syndrome (Waselenko *et al*, 2004), we examined the role of intestinal microbes on mitigating radiation-induced bone marrow toxicity and GI toxicity. The reduction in the number and/or size of the splenic germinal centre and memory CD4$^{+}$ T cells in germ-free mice (Cerf-Bensussan & Gaboriau-Routhiau, 2010) suggested an important role of gut microbiota in the development of the peripheral immune system. Likewise, we found that FMT was able to restore the size of spleens in irradiated mice, indicating that FMT might similarly strengthen the immune system of radiation-exposed mice. Owing to gut microbiota treatment elevated WBC counts in the peripheral blood slightly, we further preformed FMT combined with BMT to remedy TBI-induced toxicity. Notably, comparing with FMT or BMT alone, administration of FMT combined with BMT overtly increased the survival rate of mice after 8 Gy irradiation exposure, suggesting that bone marrow toxicity might be the main limiting factor for therapeutic effect of FMT. On the basis of our investigations, we concluded that FMT improved GI tract function and intestinal epithelial integrity after irradiation, suggesting that FMT could be a potential therapy for radiation-induced GI toxicity. In agreement with the role of

angiogenesis in radioprotection (Okunieff *et al*, 1998), we found that FMT was capable of up-regulating *Vegf* level in the small intestine of irradiated mice. Because VEGF also facilitates the development of tumorigenesis (Carmeliet & Jain, 2011), we determined whether enteric microbiota transplant could accelerate the growth of cancer cells following subcutaneous implantation of either lung or colon cancer cells into C57BL/6 mice. Our data showed that FMT failed to enhance the proliferation of the carcinoma cells. Therefore, as a potential therapeutic approach to mitigate radiation-induced toxicity, FMT might be used in tumour radiotherapy to improve the prognosis.

At the molecular level, microarray analysis revealed a marked difference in the mRNA expression profile in small intestine tissues between FMT-treated and saline-treated mice. Our observation showed that the gavage of sex-matched faecal microbiota had distinguishable effects on immunity and metabolism (Marcobal *et al*, 2013; Bromberg *et al*, 2015), suggesting that different sex-specific gut microbiota performs divergent physiologic status. Given that lncRNAs, which have been primarily studied in the context of genomic imprinting, cancer, and cell differentiation, are

now emerging as important regulators of immunity and metabolism (Cui et al, 2015a,b; Yao et al, 2016), we analysed the related lncRNA expression pattern in parallel. As expected, FMT caused an alteration in the spectrum of lncRNA expression, indicating that gut microbe transplant not only preserves intestinal bacterial composition in hosts but also retains the mRNA and lncRNA expression profile. LncRNAs are able to interact with chromatin at several thousand different locations across multiple chromosomes and govern large-scale gene expression programmes (Guttman & Rinn, 2012; Vance & Ponting, 2014), and they have been regarded as pivotal modulators of physiologic or pathologic status (Ponting et al, 2009). They also regulate gene expression profiles at the post-transcriptional level through influencing the stability of mRNA (Gong & Maquat, 2011; Kretz et al, 2013). Our findings now suggest that the gut microbe transplant-mediated lncRNA expression profile fluctuations might play important roles in mitigating radiation-induced injury, which warrants further study.

In conclusion, our work demonstrates that gavage of gut microbes alleviates and protects against radiation-induced injury in a mouse model. Specifically, FMT is able to heighten the survival rate of irradiated mice. Moreover, FMT improves GI tract function and epithelial integrity to ameliorate irradiation-induced GI toxicity. Mechanistically, intestinal microbiota transplant preserves the bacterial communities and retains mRNA as well as the lncRNA expression profile in the hosts. Thus, our findings provide new insights into the function of FMT. Clinically, our observations underpin the suggestion that FMT might be employed as a novel therapeutic method for irradiation-induced injury to improve the prognosis of patients after radiotherapy.

# Materials and Methods

### Animal studies

Six- to eight-week-old male and female C57BL/6J mice were purchased from Vital River (Beijing, China), mice were housed in the specific pathogen-free level animal facility at the Institute of Radiation Medicine (IRM), the Chinese Academy of Medical Sciences (CAMS). Mice were kept under standard conditions (ambient temperature $22 \pm 2°C$, air humidity 40–70% and a 12/12-h light/dark cycle) and continuous access to a standard diet and water. All male and female mice in this study were of a pure C57BL/6 genetic background and separated into groups randomly, treated according to the guidelines established by the National Institutes of Health Guide (NIH) for use. All experiments were done in accordance with procedures approved by the Daegu-Gyeongbuk Medical Innovation Foundation (DGMIF) Institutional Animal Care and Use Committee (IACUC). All procedures and animal handlings were performed following the ethical guidelines for animal studies.

### Irradiation studies

A Gammacell® 40 Exactor (Atomic Energy of Canada Limited, Chalk River, ON, Canada) at a dose rate of 1.0 Gy per minute was used for all experiments. Non-anaesthestized mice were immobilized in a specific steel chamber, and radiation dose was monitored by a dose

rate meter (Cui et al, 2016). Male (approximately 20 g in body weight) and female (approximately 18 g) mice were treated with a single dose of 6.5 Gy gamma ray at a rate of 1.0 Gy/min (total body irradiation, TBI). After irradiation, the mice were returned to the animal facility for daily observation and treatment as described below. The body weight of radiation-exposed mice was assessed dieb. tert.

### Antibiotics test

Weaning male and female mice (about 4-week-old) were treated for 4–6 weeks with ampicillin (1,000 μg/ml) or streptomycin (500 μg/ml) in their drinking water, respectively. The fresh antibiotic solution was prepared every day to promise its activity. Antibiotics were purchased from Sigma-Aldrich (Saint-Quentin Fallavier, France). Male (approximately 22 g) and female (approximately 20 g) mice were treated with single dose of 6.5 Gy at a rate of 1.0 Gy/min.

### Donor stool preparation and administration

The healthy 6- to 8-week-old male and female C57BL/6 mice were kept in same housing and environmental conditions. Conventional, untreated, age-matched male and female mice were used as donors to collect gut microbiota. The donor's faecal pellets were collected under SPF conditions. Donor stool was freshly prepared on the day of transplant and that in all cases was prepared and transplanted within 4 h. Donor stool was weighed and diluted with 1 ml of saline per 0.1 g of stool. Briefly, the stool was steeped in saline for about 15 min, shaken and then centrifuged at 800 rpm for 3 min. The supernatant was obtained for treatment. The mice in each group were treated with same sample. Same weight of male and female stool was mix to perform male/female gut microbe mixture.

### Peripheral blood cell counts

One hundred microlitres of peripheral blood was obtained from the orbital sinus using a micro-pipette coated with the anticoagulant, $K_2EDTA$, 15 days after 6.5 Gy irradiation. The cell counts included white blood cells (WBCs) and haemoglobin (HGB) was counted using a Celltac E hemocytometer (Nihon Kohden, Japan).

### Measure of spleens

Male (approximately 20 g) and female (approximately 18 g) C56BL/6 mice were treated with single dose of 6.5 Gy gamma ray. After 15 days, the mice were euthanized, and the spleens were excised and measured.

### Formed faecal collection

For this study, six mice (6- to 8-week-old male or female C57BL/6J mice) were fed in one cage. They were kept under the same environment with other experimental mice. Formed faecal were collected from cage bedding on the day before irradiation, day 1 after irradiation, day 5 after irradiation and day 10 after irradiation. Then, we counted the number of formed stools in both saline group and FMT group.

## Histology

At day 21 after 6.5 Gy TBI, mice were anaesthetized and the entire small intestine was harvested and used to collect tissue for histology. Briefly, small intestine tissues were fixed in 4% paraformaldehyde solution overnight, and 4-μm paraffin sections were processed to performed H&E and IHC staining. For H&E staining, the sections were dipped in haematoxylin for 10 min and eosin for 3 min (Perez-Garcia *et al*, 2015). For IHC staining, primary antibody of anti-F8 (M0616, Dako, Glostrup, Denmark) was used (Koo *et al*, 2009). Small intestine tissues were fixed in methanol-Carnoy (60% dry methanol, 30% chloroform, 10% glacial acetic acid) solution overnight, and 4-μm paraffin sections were processed to performed AB-PAS and PAS staining. For AB-PAS staining, dewaxed sections were hydrated and incubated in AB for 20 min. Sections were subsequently washed with water before incubation in 1% periodic acid for 10 min followed by incubation in Schiff's reagent for 10 min. Sections were counterstained with Mayer's haematoxylin for 30 s, washed and dehydrated before mounting with Pertex. For PAS staining, dewaxed sections were hydrated and incubated in 1% periodic acid for 10 min followed by incubation in Schiff's reagent for 10 min. Sections were counterstained with Mayer's haematoxylin for 30 s, washed and dehydrated before mounting with Pertex.

## FITC–dextran permeability experiments

The mice were administrated with (or without) sex-matched FMT following 6.5 Gy total body irradiation. After 21 days, mice were fasted 14 h and gavaged with 60 mg per 100 g body weight of fluorescein isothiocyanate–dextran (FITC–dextran, FD4, 3,000–5,000 kD, Sigma-Aldrich, Madrid, Spain) in a volume of 0.2 ml. Blood samples were obtained by cardiac puncture at 4 h after administration of FITC–dextran and centrifuged at 860 *g* for 5 min at room temperature to obtain the serum. Fluorescence intensity of each serum sample (DTX 880 Multimode Detector, Beckman Coulter, CA, USA) was measured.

## Quantitative real-time polymerase chain reaction

Total RNA was extracted from the tissues using TRIzol (Invitrogen, Carlsbad, CA, USA) according to the manufacturer's protocol. Reverse transcription was performed using poly (A)-tailed total RNA and reverse transcription primer with ImPro-IIReverse Transcriptase (Promega, Madison, WI, USA), according to the manufacturer's instructions. The Quantitative real-time polymerase chain reaction (qRT–PCR) was performed according to the instructions of Fast Start Universal SYBR Green Master (Rox) (Roche Diagnostics GmbH Mannheim, Germany). All primers are listed in Appendix Table S1. GAPDH was used as the controls.

## *In vivo* tumorigenicity assay

C57BL/6 mice were housed and treated according to the guidelines established by the National Institutes of Health Guide for the Care and Use of Laboratory Animals. We conducted animal transplantations according to the Declaration of Helsinki. Briefly, HT29 (or A549) cells were harvested and re-suspended at $2 \times 10^7$ cells per ml

in sterile phosphate-buffered saline. Groups of 4-week-old male and female C57BL/6 mice (Experiment Animal Center of Peking, China) (each group, *n* = 5) were subcutaneously injected at the shoulder with 0.2 ml of the cell suspensions and treated with faeces microorganism (or saline) for 10 days. After 20 days, tumour-bearing mice were euthanized, and the tumours were excised and measured.

## Bacterial diversity analysis

For this study, four mice were fed in one cage. Stool samples were freshly collected and stored at −80°C until use. DNA was extracted from the stool using the Power Fecal® DNA Isolation Kit (MoBio Carlsbad, CA USA). The DNA was recovered with 30 ml of buffer in the kit. The 16S ribosomal RNA (rRNA) V4 gene was analysed to evaluate the bacterial diversity using Illumina Hiseq (Novogene Bioinformatics Technology Co., Ltd.). Sequences analysis was performed by Uparse software (Uparse v7.0.1001, http://drive5.com/uparse/). Sequences with ≥ 97% similarity were assigned to the same OTUs. Representative sequence for each OTU was screened for further annotation. For each representative sequence, the Silva123 Database was used based on RDP classifier (Version 2.2, http://sourceforge.net/projects/rdp-classifier/) algorithm to annotate taxonomic information. The primers are listed in Appendix Table S1.

## Microarray analysis

Briefly, three male and female C57BL/6 mice treated with saline were grouped as controls. The gut microbes of three males used to treat male mice and the gut microbes of three females used to treat female mice were regarded as individual experimental groups. The mice were euthanized at day 21 after irradiation, and the small intestine tissues were excised. These three paired samples were used to synthesize double-stranded complementary DNA (cDNA), and the cDNA was labelled and hybridized to lncRNA + mRNA Capitalbiotech Sureprint Mouse lncRNA Gene Expression 4*180K (CapitalBio Corp, Beijing, China) according to the manufacturer's instructions. The data from the lncRNA + mRNA microarray were used to analyse the data summarization, normalization and quality control using Gene Spring software V11.5 (Agilent). The differentially expressed genes were selected if the change of the threshold values was > 2.0-fold and if the Benjamini–Hochberg-corrected *P*-values were < 0.05. The data were normalized and hierarchically clustered with CLUSTER 3.0 software. The data were visualized with Java Tree view software. GEO accession number: GSE92249.

## Statistical analysis

Each experiment was repeated at least three times. Significance was assessed by comparing the mean values (standard deviation; SD) using Student's *t*-test and Wilcoxon rank sum test for independent groups as follows: *$P$ < 0.05, **$P$ < 0.01, ***$P$ < 0.001. Kaplan–Meier analysis was performed for survival analysis, and significance between survival curves was determined by a log-rank test.

**Expanded View** for this article is available online.

**The paper explained**

**Problem**

Unexpected irradiation exposure causes serious acute radiation syndrome, such as life-threatening bone marrow and intestinal injury, urgently needing effective therapy. Moreover, radiotherapy towards malignancies inextricably intertwined with acute and late complications due to the side effects of irradiation which are the main challenge for radiation oncologists, medical physicists and radiobiologists. To date, radiation-induced toxicity remains a conundrum impairing the life quality of victims.

**Results**

Here, we identified that faecal microbiota transplantation (FMT) alleviates radiation-induced toxicity using mouse model. High-throughput sequencing showed that male and female mice harboured different intestinal bacterial composition which was associated with susceptibility to radiation toxicity. Transplantation of faecal microbiota from healthy mice to irradiated mice improved gastrointestinal tract function and epithelial integrity of small intestines in a sex-dependent fashion to ameliorate radiation-induced toxicity. Importantly, FMT preserves the enteric bacterial composition and retained RNA expression profile of radiation-exposed mice. In addition, FMT failed to facilitate the proliferation of cancer cell *in vivo*.

**Impact**

Faecal microbiota transplantation can be employed as a therapeutic to ameliorate radiation-induced toxicity and improve the prognosis of patients after radiotherapy. More important, the sex as well as the age between donors and recipients influences the efficiency of FMT.

## Acknowledgements

This work was supported by grants from the National Natural Science Foundation of China (No. 81502664, 81572969 and 81402541), CAMS Innovation Fund for Medical Sciences (CIFMS, 2016-I2M-1-017), the PUMC Youth Fund and the Fundamental Research Funds for the Central Universities (No. 33320140187, 3332016099 and 3332016143), the IRM-CAMS Research Fund (No. 1547 and 1522), the Technology and Development and Research Projects for Research Institutes, Ministry of Science and Technology (2014EG150134), the Tianjin Science and Technology Support Plan Project (TJKJZC, 14ZCZDSY00001). H.W. is supported by the U.S. National Center for Complementary and Alternative Medicine (NCCAM, R01AT005076) and the National Institute of General Medical Sciences (NIGMS, R01GM063075).

## Author contributions

MC, HX and SF designed and performed experiments, analysed data and wrote the paper. LZ, SZ, DL, YL, YZ, QZ, JD, XZ, JZ and LL performed experiments. HW wrote the paper. MC and SF oversaw the entire project, designed experiments, analysed data and wrote the paper.

## Conflict of interest

The authors declare that they have no conflict of interest.

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
