## [Review Process File · EMBO Molecular Medicine]

Manuscript EMM-2016-06932

Faecal microbiota transplantation protects against radiation-induced toxicity

Ming Cui, Huiwen Xiao, Yuan Li, Lixin Zhou, Shuyi Zhao, Dan Luo, Qisheng Zheng, Jiali Dong, Yu Zhao, Xin Zhang, Junling Zhang, Lu Lu, Haichao Wang, Saijun Fan

Corresponding author: Ming Cui & Saijun Fan, Tianjin Key Laboratory of Radiation Medicine and Molecular Nuclear Medicine, Institute of Radiation Medicine

Review timeline:

Submission date:	10 August 2016
Editorial Decision:	06 September 2016
Revision received:	06 December 2016
Editorial Decision:	09 January 2017
Revision received:	20 January 2017
Accepted:	25 January 2017

Editor: Céline Carret

Transaction Report:

1st Editorial Decision

06 September 2016

Thank you for the submission of your manuscript to EMBO Molecular Medicine. We have now heard back from the two referees whom we asked to evaluate your manuscript. Although the referees find the study to be of potential interest, they also raise a number of concerns that need to be addressed in the next final version of your article.

As you will see from the comments below, both referees find the study novel and interesting. However, both referees also recommend providing better descriptions, more details, ethical statements (this is mandatory for publication), and clarifications. The reviewer #1 also suggest adding some mechanistic insights to further improve attractiveness and robustness of the data.

It is our opinion that all suggested experiments and text modifications are reasonable and would improve the impact of the paper and I would therefore encourage you to address these in a major revision of your work. Please note that it is EMBO Molecular Medicine policy to allow only a single round of revision and that, as acceptance or rejection of the manuscript will depend on another round of review, your responses should be as complete as possible.

Revised manuscripts should be submitted within three months of a request for revision; they will otherwise be treated as new submissions, except under exceptional circumstances in which a short extension is obtained from the editor. EMBO Molecular Medicine has a "scooping protection" policy, whereby similar findings that are published by others during review or revision are not a

criterion for rejection. Should you decide to submit a revised version, I do ask that you get in touch after three months if you have not completed it, to update us on the status.

Please read below for important editorial formatting and amendments for submitting the next version of your article.

I look forward to receiving your revised manuscript.

***** Reviewer's comments *****

Referee #1 (Comments on Novelty/Model System):

In this study, Cui and colleagues sought to examine the impact of fecal microbiota transplantation (FMT) on the host following total body irradiation. The authors demonstrate that administration of feces from healthy donors to irradiated recipients reduces radiation-associated mortality in a sex-dependent fashion as a result of changes in intestinal microbial communities. In addition, FMT administration was shown to change the expression level of genes involved in immunity and metabolism. Finally, the authors show that tumor development is not affected by FMT administration. All together these observations suggest the potential for FMT in ameliorating the toxic side effects associated with total body irradiation in cancer patients.

Overall evaluation:

Total body irradiation causes damage to the intestinal lining leading to intestinal permeability and translocation of opportunistic pathogens into the bloodstream. As a matter of fact, prophylactic antibiotic treatment is used in the clinical setting to prevent infection-associated complications. Therefore, administration of fecal bacteria soon following irradiation would seem counterintuitive. Nevertheless, the data shown in this report demonstrates not only that transplantation of feces ameliorates the deleterious side effects of radiation but also that this beneficial effect is influenced by the sex of donor and recipient. The main problem with the study is that it is descriptive in nature. Although differences in microbial composition and gene expression may explain the protective phenotype, it is not fully clear how FMT protects mice from radiation-induced toxicity. Recovery of epithelial integrity is mentioned but not addressed experimentally. Despite this, the findings presented here are novel and of interest to the field.

Specific comments:

1. In Fig. 1, a more detailed comparison of the bacterial composition differences in male and female microbiota pre and post radiation is important. Measurement of biodiversity (Shannon/Inverse Simpson index) as well as composition bar plots should be included to better demonstrate the differences between male and female microbiota and the changes induced by total body irradiation.
2. In Fig. 2A, is the data shown before or after TBI? If post TBI, how does this compare to pre-irradiation? Also in Fig. 2A, the authors claim that male mice harbor more Bacteroides and lower Firmicutes compared to female animals. This is difficult to tell from the figure since bacterial taxa is shown at the genus as opposed to phylum level. It would be helpful to group genera into their corresponding phyla for clarification.
3. FMT is administered to mice in the absence of antibiotic treatment. In addition to changes in bacterial composition, does irradiation reduce the total bacterial density? How stable is engraftment of FMT organisms?
4. The claim that FMT is restoring epithelial integrity after irradiation needs to be explored further. Is FMT helping with intestinal permeability? This is mentioned but not addressed experimentally. To assess intestinal permeability one could measure FITC dextran levels in control and FMT-treated mice. Additionally, changes in expression of Muc2 genes, mucus layer thickness and number of goblet cells post FMT would make a stronger argument in favor of a mechanism involving the restoration of epithelial integrity.

5. Irradiation leads to weight loss and diarrhea. How does the weight gain occurring post FMT in Fig. 2 compare to that of non-irradiated animals? Does FMT fully restore pre-TBI weight?
6. In Figures 4, EV3, 5 E-H and S4 data should be presented in chronological order (0, 5, 10 days)
7. Data from Fig 4 and Fig. EV3 suggest that FMT from male or female donors do not fully engraft in sex-mismatched recipients. This is an important point because the differences in survival following FMT could be explained by failure of key bacterial strains in male and female mice to engraft in the recipients as opposed to failure to respond to FMT despite efficient engraftment. To what extent does the microbiota of transplanted animals resemble the input (donor material)? It would be helpful to show the % of taxa that colonized following sex-matched and sex-mismatched FMT in all recipients.
8. In Fig 6C-D and E-F, what does C and T stand for? This needs to be indicated in the figure legend. What time point does the microarray analysis correspond to? The authors say that FMT re-established the intestinal bacterial composition and reprogrammed mRNA and long non-coding RNA expression profiles of host small intestines in a sex-specific fashion. However, the gene expression profile in the sex-mismatched FMT group is not shown.
9. In Figures 7D and E, bar graphs are missing error bars.

Referee #1 (Remarks):

This study provides additional evidence on the beneficial role of FMT in the clinic. Please refer to the comments above for comments regarding the structure of the data shown plus additional necessary information to strengthen the authors' claims.

Referee #2 (Comments on Novelty/Model System):

The studies have been done thoroughly but the descriptions, in particular of methods, lack enough detail for the reader to be able to reproduce them.

The study uses faecal microbial transplant as a means to ameliorate gut and bone marrow toxicity due to irradiation. These toxicities are a significant dose-limiting problem of radiotherapy and chemotherapy and possible therapies are much sought after. I am not aware that the faecal microbial transplant procedure has been tried before in this context.

The experimental model used was appropriate, but the description and procedures lack detail. Although there does not appear to be any ethical issues with the study, there is no specific statement that the studies were ethically reviewed and approved in accordance with national regulations.

The findings are novel and important, but the paper needs considerable revision before consideration for publication.

Referee #2 (Remarks):

Manuscript Number: EMM-2016-06932

Title: Faecal microbiota transplantation protects against radiation-induced toxicity Corresponding Author: Dr. Cui

This is a very interesting piece of work with some very important findings with regard to the potential use of faecal microbiota transplant (FMT) to ameliorate gut and bone marrow toxicity caused by irradiation. Whilst FMT was protective against gut toxicity, the effect on bone marrow toxicity are however equivocal. Also, the protection may have been due to viable bacteria in the FMT, bioactive factors in the supernatant or both.

Pg 4 para 1 ln 14-15 "enteric bacteria and the radio-sensitivity"
Should consider including reference to work by research groups of Keefe and Sonis in this area.

Vanhoecke B, De Ryck T, Stringer A, Van de Wiele T, Keefe D. Microbiota and their role in the pathogenesis of oral mucositis. *Oral Dis*. 2015 Jan;21(1):17-30. doi: 10.1111/odi.12224. Epub 2014 Feb 25. PubMed PMID: 24456144.

Vasconcelos RM, Sanfilippo N, Paster BJ, Kerr AR, Li Y, Ramalho L, Queiroz EL, Smith B, Sonis ST, Corby PM. Host-Microbiome Cross-talk in Oral Mucositis. *J Dent Res*. 2016 Jul;95(7):725-33. doi: 10.1177/0022034516641890. Epub 2016 Apr 6. Review. PubMed PMID: 27053118; PubMed Central PMCID: PMC4914867.

Pg 5 para 2 "Intestinal bacteria"

Mice were given antibiotic for 6 weeks prior to dosing. The assumption is that the treatment has changed the composition of the microbiota but this is not shown. Although a reasonable assumption, it would be useful to show its nature, since the antibiotic treatments affected gut toxicity but not bone marrow toxicity whereas faecal transplant is suggested to protect against both.

Was antibiotic treatment only prior to irradiation or was it continued post-irradiation?

Pg 6 para 1 ln 5 "stool of male and female were different"

How much inter-animal variation was there?

Faecal donors were older. Did the composition of the flora change with time?

Were donors kept in same housing and environmental conditions? Again, this can have a major effect on microbiota.

Pg 7 para 1 "spleen, WBC"

Interesting that spleen protection occurred equally irrespective of faecal source.

WBC increased above TBI alone, but still marginal protection compared to levels in un-treated.

Suggests that FMT had quite limited effects on bone marrow toxicity, as noted with antibiotics. In discussion it is indicated that whilst FMT prolonged mouse survival it did not prevent their demise.

Was this due to the bone marrow toxicity?

Pg 7 para 2 ln 8-12 "genes"

The expression of these genes are far higher than with TBI alone or controls. Does this indicate an active repair or is there an ongoing problem in the gut despite the improved morphology.

Pg 8 para 1

The changes in the microbiota vary with time, with potentially protective changes occurring by 5 days but declining thereafter. This possibly should be discussed more, since it may indicate a critical but short term period in which the treatment is protective.

Pg 9 para 1 ln 7 "recuperative"

Cannot really say that flora of female mice was more recuperative since they did not survive any better than the males. It may be that it has fewer detrimental bacteria in the first instance.

Interesting that effects of FMT, had more pronounced effects on WBC in females than in males.

Pg 9 para 1 ln 11 "re-establishes"

It could be that FMT helps preserve the original gut microbiota rather than re-establishes it.

Pg 10 para 1 ln 7 "reprogrammed"

Again, it may be retained rather than re-programmed.

Pg 12 para 2 ln 6-8 "age of donor and receipts"

Not sure what is meant in this sentence, other than that FMT prolonged mouse survival but did not prevent their demise. This information should be indicated in results, rather than as an add-on in the discussion.

Pg 13 para 2 ln 9 "WBC counts"

Protection due to FMT was limited.

Pg 14 para 1 ln 7-8 "female mice exhibit higher irradiation tolerance than males"

Data suggest females have either have similar sensitivity to radiation or are more susceptible.

Pg 15 para 3 "Animal studies"

What was source and age of the mice? Were mice bred on site or how was it ensured that donors were from same parents? What were the housing and environmental conditions for the mice? Were the experiments subject to ethical and welfare review by regulatory authorities?

Did non-survivors die during experiment, were they euthanased due to loss of health and condition or a combination of both? What health and welfare criteria were used?

Pg 16 para 2 "irradiation studies"

Need more detail as to how treatment conditions established or a published reference to them.

Pg 16 para 3 "antibiotics"

Were antibiotics given only prior to irradiation or prior to and post irradiation.

How often were the antibiotic solution replaced and were water intakes by mice similar for all groups?

Pg 16 para 4 "Donor stool"

This description needs far more detail, since it is fundamental to the paper.

Looks that procedure done aerobically. Many anaerobes would not have survived these conditions.

How viable was the suspension? Important, because protective effects may be due to viable microbes in the FMT but could also be a result of microbe-derived bioactive factors in supernatant.

Need detail on nature and age of the donor mice. Also, how were faeces samples collected?

Pg 17 para 5 "H & E"

Need more detail on tissue collection, processing and cutting or a reference detailing the procedures used.

For most of the methods used citing of a literature reference would be helpful for the reader.

Figures

These should be stand alone, without need to refer to main text

Figure 1 E, F

Figure 2 G, H

When were WBC counts done (15d)?

Figure 2 E, F Spleen dissected when (21d)?

Figure 3 A, D-G When?

Figure 4 C, D & G, H If emphasis is on top 10, possibly take "others" out of plot to allow differences amongst the 10 to show more clearly. As is the "others" look to be the more important component of the microbiota

Figure 6 and 7 When samples taken?

Similar general comments for expanded view figures

1st Revision - authors' response

06 December 2016

Reply to Referee #1' comments:

Specific comments:

Question 1: In Fig. 1, a more detailed comparison of the bacterial composition differences in male and female microbiota pre and post radiation is important. Measurement of biodiversity (Shannon/Inverse Simpson index) as well as composition bar plots should be included to better

demonstrate the differences between male and female microbiota and the changes induced by total body irradiation.

Answer: Thanks for the suggestion. Biodiversity (Observed species and Shannon index) was measured to better represent the alteration of gut microbiota induced by TBI. Intriguingly, significant changes of gut microbiota was observed in male mice ($p=0.01394$ and 0.007957), not in female mice ($p=0.4460$ and 0.2083) 10 day after TBI, as shown in Fig 1A and B. A significant difference of gut bacterial between male and female mice were observed ($p=0.0381$ and 0.01446), as shown in Fig 2A.

Question 2: In Fig. 2A, is the data shown before or after TBI? If post TBI, how does this compare to pre-irradiation? Also in Fig. 2A, the authors claim that male mice harbor more Bacteroides and lower Firmicutes compared to female animals. This is difficult to tell from the figure since bacterial taxa is shown at the genus as opposed to phylum level. It would be helpful to group genera into their corresponding phyla for clarification.

Answer: Thanks for the suggestion. The samples with gut bacterial composition between male and female mice were collected before irradiation, the description was modified in our revised manuscript. Ten of the most abundant enteric bacteria between male and female mice were detected at the genus level, a higher frequency of Bacteroides and a lower frequency of Prevotella were observed in male mice comparing with female mice, as shown in Fig 2B. Writing was modified in our revised manuscript.

Question 3: FMT is administered to mice in the absence of antibiotic treatment. In addition to changes in bacterial composition, does irradiation reduce the total bacterial density? How stable is engraftment of FMT organisms?

Answer: Thanks for the suggestion. The total gut bacterial density of male mice was decreased at 3 day after TBI in our recent studies (Cui M et al., Int. J. Mol. Sci. 2016, 17, 1786; doi:10.3390/ijms17111786). The total gut bacterial density of male mice unchanged at day 5 and increased at day 10 after TBI in our present studies. We also found that the total gut bacterial density of male mice was increased at day 7 after TAI (unpublished data), suggesting that exposure to ionizing radiation decreases the total gut bacterial density of male mice in short-term, but increases the total gut bacterial density in long-term. However, no significant alterations with the total gut bacterial density was seen in female mice irradiated under the same condition, as shown in Fig 1A and B. Beta diversity analysis revealed that FMT shaped the radiation-changed gut bacterial composition both in male and female mice, indicating that engraftment of FMT organisms is stable in the system (Fig 5A-D and EV4A-D).

Question 4: The claim that FMT is restoring epithelial integrity after irradiation needs to be explored further. Is FMT helping with intestinal permeability? This is mentioned but not addressed experimentally. To assess intestinal permeability one could measure FITC dextran levels in control and FMT-treated mice. Additionally, changes in expression of Muc2 genes, mucus layer thickness and number of goblet cells post FMT would make a stronger argument in favor of a mechanism involving the restoration of epithelial integrity.

Answer: Thanks for the suggestion. As FITC-dextran was used for assay of *in vivo* intestinal permeability, FMT significantly decreased or inhibited the radiation-increased FITC-dextran levels in peripheral blood of mice, suggesting that FMT is helpful for intestinal permeability (Fig 3D and EV2D). A series of experiments were conducted for the expression level of *Muc2* in small intestine by qRT-PCR, the mucus layer thickness by AB-PAS staining and the number of goblet cells by PAS staining. As shown in Figs 3A&E and EV2A&E, FMT thickened the mucus layer, increased the number of goblet cells and up-regulated *Muc2* mRNA expression in male and female mice, providing further evidence that FMT might mitigate radiation-induced GI toxicity.

Question 5: Irradiation leads to weight loss and diarrhea. How does the weight gain occurring post FMT in Fig. 2 compare to that of non-irradiated animals? Does FMT fully restore pre-TBI weight?

Answer: Thanks for the suggestion. As observed, FMT significantly increased the number of formed feces, helped with the intestinal permeability and improved the epithelial integrity after irradiation, suggesting that FMT could meliorate radiation-affected GI tract function resulting in the weight gain of irradiated mice. Four groups, i.e., mice treated with saline, mice treated with sex-match FMT, mice treated with sex-mismatch FMT and sex-mixed FMT were used in our studies. All the three kinds of FMT increased weight of irradiated mice, which were lower than that of mice pre-TBI.

Question 6: In Figures 4, EV3, 5 E-H and S4 data should be presented in chronological order (0, 5, 10 days)

Answer: Thanks for the suggestion. Modification was done in Figs 4, 5, EV3 and EV4.

Question 7: Data from Fig 4 and Fig. EV3 suggest that FMT from male or female donors do not fully engraft in sex-mismatched recipients. This is an important point because the differences in survival following FMT could be explained by failure of key bacterial strains in male and female mice to engraft in the recipients as opposed to failure to respond to FMT despite efficient engraftment. To what extent does the microbiota of transplanted animals resemble the input (donor material)? It would be helpful to show the % of taxa that colonized following sex-matched and sex-mismatched FMT in all recipients.

Answer: Thanks for the suggestion. As shown by Beta diversity analysis (Figs 5A&B and EV4A&B), irradiation significantly altered gut bacterial composition in male mice, sex-matched, sex-mismatched and sex-mixed FMT shaped the radiation-changed gut bacterial composition overtly, suggesting that the three kinds of engraftments were efficient. However, the frequency of specific bacteria changed by the three kinds of FMT at genus level was different in male mice. For instance, a decrease of [prevotella] frequency in mice and an increase in irradiated mice supplied with sex-matched and sex-mixed FMT were observed. However, sex-mismatched FMT failed to increase the frequency of [prevotella], suggesting that the sex of donor influences the gut bacterial composition of recipient following FMT (Figs. 4A&B and EV3A&B).

In female mice, the three kinds of engraftments were also efficient (Figs. 5C&D and EV4C&D), and at genus level, the frequency of specific bacteria changed by the three kinds of FMT was different (Figs. 4C&D and EV3C&D). Writing modification was provided in our revised manuscript.

Question 8: In Fig 6C-D and E-F, what does C and T stand for? This needs to be indicated in the figure legend. What time point does the microarray analysis correspond to? The authors say that FMT re-established the intestinal bacterial composition and reprogrammed mRNA and long non-coding RNA expression profiles of host small intestines in a sex-specific fashion. However, the gene expression profile in the sex-mismatched FMT group is not shown.

Answer: Thanks for the suggestion. In Fig 6C-D and E-F, "C" stands for mice treated with saline after irradiation, "T" stands for mice treated with sex-matched FMT after irradiation, "TBI+Saline" and "TBI+sex-matched FMT" to replace "C" and "T" in Fig 6 in our revised manuscript. We also performed microarray analysis to assess the gene expression profile both in mice with the sex-mismatched and sex-mixed FMT, and found the changes of gene expression profile mediated by different kinds of FMT were various (Figs S2 and S3). Writing modification was provided in our revised manuscript.

Question 9: In Figures 7D and E, bar graphs are missing error bars.

Answer: The error bars were provided in modified Figs 7D and F.

Reply to Referee #2' comments:

Question 1: Pg 4 para 1 ln 14-15 "enteric bacteria and the radio-sensitivity"

Should consider including reference to work by research groups of Keefe and Sonis in this area. Vanhoecke B, De Ryck T, Stringer A, Van de Wiele T, Keefe D. Microbiota and their role in the pathogenesis of oral mucositis. Oral Dis. 2015 Jan;21(1):17-30. doi: 10.1111/odi.12224. Epub 2014 Feb 25. PubMed PMID: 24456144.

Vasconcelos RM, Sanfilippo N, Paster BJ, Kerr AR, Li Y, Ramalho L, Queiroz EL, Smith B, Sonis ST, Corby PM. Host-Microbiome Cross-talk in Oral Mucositis. J Dent Res. 2016 Jul;95(7):725-33. doi: 10.1177/0022034516641890. Epub 2016 Apr 6. Review. PubMed PMID: 27053118; PubMed Central PMCID: PMC4914867.

Answer: Thanks for the suggestion. References were provided in our revised manuscript.

Question 2: Pg 5 para 2 "Intestinal bacteria"

Mice were given antibiotic for 6 weeks prior to dosing. The assumption is that the treatment has changed the composition of the microbiota but this is not shown. Although a reasonable assumption, it would be useful to show its nature, since the antibiotic treatments affected gut toxicity but not bone marrow toxicity whereas faecal transplant is suggested to protect against both.

Was antibiotic treatment only prior to irradiation or was it continued post-irradiation?

Answer: Thanks for the suggestion. Assay of gut microbiota alteration in male and female mice before and after ampicillin (or streptomycin) treatment was conducted. As expected, exposure to antibiotics significantly changed the gut bacterial composition (Fig 1C and D). Writing modification was provided in our revised manuscript.

Question 3: Pg 6 para 1 ln 5 "stool of male and female were different"

How much inter-animal variation was there?

Faecal donors were older. Did the composition of the flora change with time?

Were donors kept in same housing and environmental conditions? Again, this can have a major effect on microbiota.

Answer: Thanks for the suggestion. No significant individual difference among male (or female) mice was observed as heatmap analysis (Appendix Fig S1E), but significant difference between male and female mice was seen by Shannon index and Beta diversity analysis (Fig 2A). No change of the flora composition for mice housed in the Specific Pathogen Free level was seen during one month (Appendix Fig S1A and B). Thus, the composition of flora from donors did not change in our experiment. And the donors were housed in same housing and environmental conditions during the experiment.

Question 4: Pg 7 para 1 "spleen, WBC"

Interesting that spleen protection occurred equally irrespective of faecal source.

WBC increased above TBI alone, but still marginal protection compared to levels in un-treated.

Suggests that FMT had quite limited effects on bone marrow toxicity, as noted with antibiotics. In discussion it is indicated that whilst FMT prolonged mouse survival it did not prevent their demise.

Was this due to the bone marrow toxicity?

Answer: Thanks for the suggestion. To nail the issue, we separated the male mice into four cohorts. The mice in Group A were treated with saline as negative control; the mice in Group B were treated with sex-matched FMT; the mice in Group C were treated with bone marrow transplantation (BMT) and the mice in Group D were treated with sex-matched FMT combined with BMT. After 8 Gy irradiation, the survival rate of mice from Group D (approximately 50%) was obviously higher than the other three groups (0%). The similar results were obtained in female mice. These findings (Fig 2H, I and EV11) suggest that although FMT elevated WBC counts in peripheral blood, bone marrow toxicity might be the main limiting factor for therapeutic activity of FMT. New description was provided in our revised manuscript.

Question 5: Pg 7 para 2 ln 8-12 "genes"

The expression of these genes are far higher than with TBI alone or controls. Does this indicate an active repair or is there an ongoing problem in the gut despite the improved morphology.

Answer: Thanks for the suggestion. It has been reported that radiation can stabilize the levels of HIF1 and HIF2 expression in the intestine, which modulates the genes required for intestinal barrier function, such as TFF3 and MDR1. Moreover, DMOG increases the expression of HIF to protect against radiation-induced GI toxicity and death (Taniguchi CM et al., Sci Transl Med 6: 236ra264). The consistent results were obtained in our present studies, radiation alone increased the expression of these genes slightly, FMT overtly up-regulated those genes resulting in the improved morphology finally.

Question 6: Pg 8 para 1

The changes in the microbiota vary with time, with potentially protective changes occurring by 5 days but declining thereafter. This possibly should be discussed more, since it may indicate a critical but short term period in which the treatment is protective.

Answer: Thanks for the suggestion. Writing was modified in our revised manuscript.

Question 7: Pg 9 para 1 ln 7 "recuperative"

Cannot really say that flora of female mice was more recuperative since they did not survive any better than the males. It may be that it has fewer detrimental bacteria in the first instance.

Interesting that effects of FMT, had more pronounced effects on WBC in females than in males.

Answer: Thanks for the suggestion. Writing was modified in our revised manuscript.

Question 8: Pg 9 para 1 ln 11 "re-establishes"

It could be that FMT helps preserve the original gut microbiota rather than re-establishes it.

Answer: Thanks for the suggestion. "Re-establish" was replaced with "preserve" in our revised manuscript.

Question 9: Pg 10 para 1 ln 7 "reprogrammed"

Again, it may be retained rather than re-programmed.

Answer: Thanks for the suggestion. "Reprogramme" was replaced with "retain" in our revised manuscript.

Question 10: Pg 12 para 2 ln 6-8 "age of donor and receipts"

Not sure what is meant in this sentence, other than that FMT prolonged mouse survival but did not prevent their demise. This information should be indicated in results, rather than as an add-on in the discussion.

Answer: Thanks for the suggestion. Writing was modified in our revised manuscript.

Question 11: Pg 13 para 2 ln 9 "WBC counts"

Protection due to FMT was limited.

Answer: Thanks for the suggestion. Writing was modified in our revised manuscript..

Question 12: Pg 14 para 1 ln 7-8 "female mice exhibit higher irradiation tolerance than males"

Data suggest females have either have similar sensitivity to radiation or are more susceptible.

Answer: Thanks for the suggestion. Writing was modified in our revised manuscript.

Question 13: Pg 15 para 3 "Animal studies"

What was source and age of the mice? Were mice bred on site or how was it ensured that donors were from same parents? What were the housing and environmental conditions for the mice? Were the experiments subject to ethical and welfare review by regulatory authorities?

Did non-survivors die during experiment? Were they euthanized due to loss of health and condition or a combination of both? What health and welfare criteria were used?

Answer: Thanks for the suggestion. More information was provided in the methods section of our revised manuscript.

Question 14: Pg 16 para 2 "irradiation studies"

Need more detail as to how treatment conditions established or a published reference to them.

Answer: Thanks for the suggestion. More information was provided in the methods section of our revised manuscript.

Question 15: Pg 16 para 3 "antibiotics"

Were antibiotics given only prior to irradiation or prior to and post-irradiation.

How often was the antibiotic solution replaced and were water intakes by mice similar for all groups?

Answer: Thanks for the suggestion. More information was provided in the methods section of our revised manuscript.

Question 16: Pg 16 para 4 "Donor stool"

This description needs far more detail, since it is fundamental to the paper.

Looks that procedure done aerobically. Many anaerobes would not have survived these conditions.

How viable was the suspension? Important, because protective effects may be due to viable microbes in the FMT but could also be a result of microbe-derived bioactive factors in supernatant. Need detail on nature and age of the donor mice. Also, how were faeces samples collected?

Answer: Thanks for the suggestion. More information was provided in our revised manuscript.

Faecal microbiota transplantation has widely been performed in clinical trials (Sbahi H, BMJ Open Gastroenterol. 2016 May 9;3(1):e000087). At present, there is no standardized fecal preparation protocol for FMT. Feces are usually collected on the day of transplantation from the donor. The feces are dissolved in normal saline or water, homogenized, and filtered to make a liquid slurry as described in previous studies (Matsuoka K, Keio J Med. 2014;63(4):69-74). Normal saline is presumed to be less likely to affect the microbiota of donor stool (Choi HH, Clin Endosc. 2016 May;49(3):257-65). Accordingly, we used saline to prepare fecal materials. More importantly, faecal microbiota transplant has received increased scrutiny after numerous studies proved that stool is a biologically active complex mixture of living organisms with therapeutic potential, and the intestinal microbiota was recognized as the biologically active component of stool (Aroniadis OC,

Curr Opin Gastroenterol 2013;29:79–84). In our present study, fecal samples for FMT were collected from conventional, untreated, age-matched male and female mice. More information was provided in the methods section of our revised manuscript.

Question 17: Pg 17 para 5 "H & E"

Need more detail on tissue collection, processing and cutting or a reference detailing the procedures used.

For most of the methods used citing of a literature reference would be helpful for the reader.

Answer: Thanks for the suggestion. More information was provided in the methods section of our revised manuscript.

Question 18: Figures

These should be stand alone, without need to refer to main text

Figure 1 E, F

Figure 2 G, H

When were WBC counts done (15d)?

Figure 2 E, F Spleen dissected when (21d)?

Figure 3 A, D-G When?

Figure 4 C, D & G, H If emphasis is on top 10, possibly take "others" out of plot to allow differences amongst the 10 to show more clearly. As is the "others" look to be the more important component of the microbiota

Figure 6 and 7 When samples taken?

Similar general comments for expanded view figures

Answer: Thanks for the suggestion. Modification was provided in our revised manuscript.

2nd Editorial Decision

09 January 2017

Thank you for the submission of your revised manuscript to EMBO Molecular Medicine. We have now received the enclosed reports from the referees that were asked to re-assess it. As you will see the reviewers are now globally supportive and I am pleased to inform you that we will be able to accept your manuscript pending the following final amendments:

1) Please address the minor changes commented by the referees. Please provide a letter INCLUDING the reviewer's reports and your detailed responses to their comments (as Word file).

Please submit your revised manuscript within two weeks. I look forward to seeing a revised form of your manuscript as soon as possible.

***** Reviewer's comments *****

Referee #1 (Remarks):

The authors of Fecal Microbiota Transplantation Protects Against Radiation-Induced Toxicity did a good job addressing the reviewer's comments. However, there are some additional points that need to be addressed prior to publication of this manuscript.

1. Timepoints for data analysis, tissue collection, WBC count, etc. need to be included in the Figure Legends for each figure, including supplementary material.

2. A more detailed description of bacterial composition analyses should be included in the Methods section. What database was used to classify sequences? Were the sequences classified based on \geq 97% similarity? This is key information that needs to be provided given the emphasis on bacterial composition and diversity.

3. On page 9, paragraph 2 the authors write: "Surprisingly, the intestinal bacterial flora was more stable in female mice after radiation exposure..." This conclusion is not supported by data shown in Fig. 4E where the bacterial composition in female mice looks very different on day 5 and 10 post irradiation compared to pre-treatment (day 0). Even if the diversity scores remain unchanged due to

shifts in bacterial populations the microbiota does undergo changes and therefore, it is not stable. Please adjust the wording in this paragraph and in the Discussion section.

4. On page 13, end of paragraph 1 the authors write: "More importantly, we obtained that FMT increased the survival rate of irradiated mice, suggesting that ten times of FMT (short term therapy) might achieve long-term healing toward radiation-induced toxicity". Something along the lines of "We observed that multiple FMT doses increased the survival rate of irradiated mice suggesting that FMT administration may provide protection against radiation-induced toxicity" provides a clearer message.

5. Manuscript should be reviewed for typos. For instance, in Figure EV2 legend:

- a. Feces-microorganism transplant = Fecal microbiota transplant
- b. Meliorates = ameliorates
- c. One cohort was treated saline = One cohort was treated with saline

6. Throughout manuscript, change "dropping counts" to "fecal pellet counts"

Referee #2 (Comments on Novelty/Model System):

The study addresses a very important health problem associated with dose-limiting toxicities associated radiotherapy and indeed chemotherapy and does highlight the potential of faecal transplant to limit counteract these problems. However, the weakness is that the composition and viability of the transplant material is not defined and indeed viabil and composition may be altered by the collection / dispersal procedure since it appears to be done aerobically. The procedure used by the authors for preparation of the faecal transplant is in line with that used generally and thus acceptable. However, the composition of administered faecal transplant is undefined, in particular it is unclear what will have happened to anaerobes which are often associated with anti-inflammatory or protective actions. With the modifications that have been done, the paper is suitable for publication. However, wider use and further study of this procedure for treatment of radiation-associated toxicities will need use of more rigorously defined product.

Referee #2 (Remarks):

Question 18 Figures: information on sample collection time-points are still not clearly defined.

2nd Revision - authors' response

20 January 2017

Referee #1 (Remarks):

The authors of Fecal Microbiota Transplantation Protects Against Radiation-Induced Toxicity did a good job addressing the reviewer's comments. However, there are some additional points that need to be addressed prior to publication of this manuscript.

Question 1. Time points for data analysis, tissue collection, WBC count, etc. need to be included in the Figure Legends for each figure, including supplementary material.

Answer: Thank you for the suggestion. We added the time points for data analysis, tissue collection, WBC count, etc. in the Figure Legends, covering the supplementary material. Writing modification was provided in our revised manuscript.

Question 2. A more detailed description of bacterial composition analyses should be included in the Methods section. What database was used to classify sequences? Were the sequences classified based on $\geq 97\%$ similarity? This is key information that needs to be provided given the emphasis on bacterial composition and diversity.

Answer: Thank you for the suggestion. We rewrote the description of bacterial composition analyses and added more details in the Methods section. Writing modification was provided in our revised manuscript.

Question 3. On page 9, paragraph 2 the authors write: "Surprisingly, the intestinal bacterial flora was more stable in female mice after radiation exposure..." This conclusion is not supported by data shown in Fig. 4E where the bacterial composition in female mice looks very different on day 5 and 10 post irradiation compared to pre-treatment (day 0). Even if the diversity scores remain unchanged due to shifts in bacterial populations the microbiota does undergo changes and therefore, it is not stable. Please adjust the wording in this paragraph and in the Discussion section.
 Answer: Thank you for the suggestion. We rewrote the sentences in the paragraph and modified the Discussion section. Writing modification was provided in our revised manuscript.

Question 4. On page 13, end of paragraph 1 the authors write: "More importantly, we obtained that FMT increased the survival rate of irradiated mice, suggesting that ten times of FMT (short term therapy) might achieve long-term healing toward radiation-induced toxicity". Something along the lines of "We observed that multiple FMT doses increased the survival rate of irradiated mice suggesting that FMT administration may provide protection against radiation-induced toxicity" provides a more clear message.
 Answer: Thank you for the suggestion. We rewrote the sentence in this paragraph. Writing modification was provided in our revised manuscript.

Question 5. Manuscript should be reviewed for typos. For instance, in Figure EV2 legend:

- a. Feces-microorganism transplant = Fecal microbiota transplant*
- b. Meliorates = ameliorates*
- c. One cohort was treated saline = One cohort was treated with saline*

Answer: Thank you for the suggestion. We checked throughout the manuscript and corrected all the typos. Writing modification was provided in our revised manuscript.

Question 6. Throughout manuscript, change "dropping counts" to "fecal pellet counts"

Answer: Thank you for the suggestion. We replaced "dropping counts" to "faecal pellet counts" throughout the manuscript. Writing modification was provided in our revised manuscript.

Referee #2 (Remarks):

Question 18 Figures: information on sample collection time-points are still not clearly defined.

Answer: Thank you for the suggestion. More information was provided in the methods section of our revised manuscript.

Corresponding Author Name: Ming Cui and Saijun Fan

Manuscript Number: EMM-2016-06932-V2